

# Multi-decadal Floodplain Classification and Trend Analysis in the Upper Columbia River Valley, British Columbia

Italo Sampaio Rodrigues [1]; Christopher Hopkinson [1]; Laura Chasmer [1]; Ryan J. MacDonald [1]; Suzanne E. Bayley [2], Brian Brisco [3†]

[1]Department of Geography and Environment, University of Lethbridge, Canada
[2]Department of Biological Sciences, University of Alberta, Canada
[3]Canada Centre for Mapping and Earth Observation, Ottawa, Canada
[†] Deceased

*Correspondence to*: Italo S. Rodrigues (italo.rodrigues@uleth.ca)

**Abstract.** Floodplain wetland ecosystems experience significant seasonal water fluctuation over the year, resulting in a dynamic hydroperiod, with a range of vegetation community responses. This paper assesses trends and changes in landcover and hydro-climatological variables, including air temperature, river discharge, and water level in the Upper Columbia River Wetlands (UCRW), British Columbia, Canada. A time series landcover classification from the Landsat image archive was generated using a Random Forest algorithm from 1984 to 2022. Peak river flow timing, duration, and anomalies were examined to evaluate temporal coincidence with observed landcover trends. The land cover classifier used to segment changes in wetland area and open water performed well (Kappa = 0.82). Over the last four decades, observed river discharge and air temperature have increased, precipitation has decreased, the timing of peak flow is earlier, and flow duration has been reduced. The frequency of both high discharge events and dry years have increased, indicating a shift towards more extreme floodplain flow behavior. These hydrometeorological changes are associated with a shift in the timing of snow melt from April to mid-May and are associated with seasonal changes in the vegetative communities over the 39-year period. The area of woody shrub landcover has increased in the spring (April to mid-May), peak flow period (late-May to July) and early fall (August to mid-September) by +6% to +12% since 1984. In the spring and early fall, the area of open water has decreased –3% to –6% since 1984, while it has increased 3% during the peak flow period. The area of marsh land cover (mostly bulrush and cattails) has declined in every season by –29% in spring, –19% in the peak flow period and –5% in early fall. These findings suggest that increasing temperatures have already impacted regional hydrology, wetland hydroperiod and floodplain landcover in the Upper Columbia Valley in Canada. Overall, there is substantial variation in seasonal and annual land cover reflecting the dynamic nature of floodplain wetlands, but the results show that the wetlands are drying out with increasing the areas of woody/shrubby habitat and loss of aquatic habitat. The results suggest that floodplain wetlands, particularly marsh and open water habitats are vulnerable to climatic and hydrological changes that could further reduce their areal extent in the future.

Keywords: Montane; Floodplain Wetlands; Hydrology; Climate Change; Landsat; Landcover Classification; Seasonal Change





*Graphic Abstract*

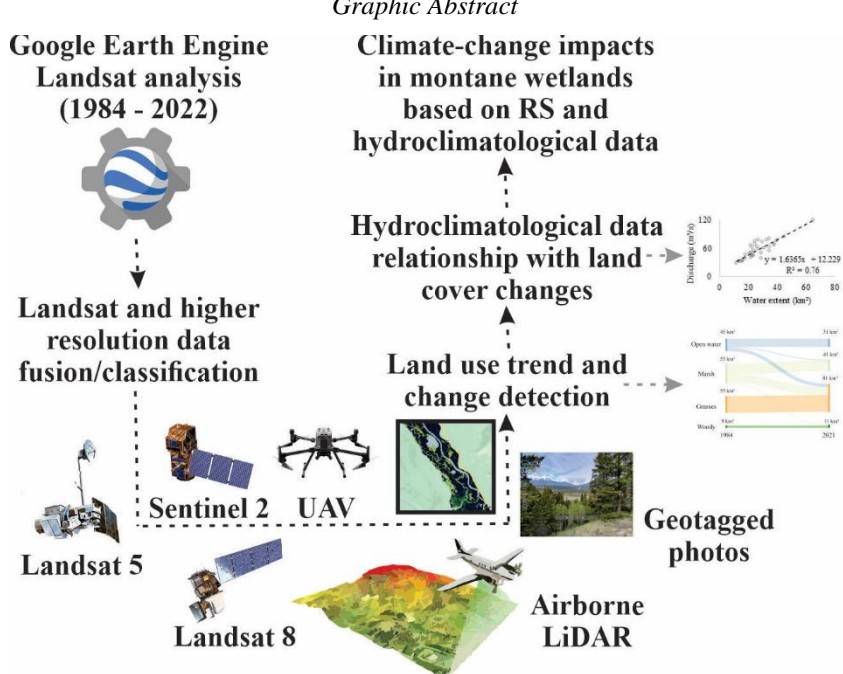

## 1. Introduction

Many montane wetlands have a short history of establishment due to the short period since the deglaciation of lower elevation areas (Cooper et al. 2012) and areminerotrophic, making them highly sensitive to changes in surface and ground-water hydrology (Hathaway et al., 2022; Wang et al., 2016; Wang et al., 2018). Large climatic gradients occur within relatively short distances due to elevational changes, which can amplify the effects of climate (MacDonald et al., 1993; Hopkinson and Young, 1998; Loeffler et al., 2011; McCaffrey and Hopkinson, 2020).

Over the last several decades, climatic changes and the amplifying effects of large elevational gradients on micro-climatology in the Canadian Rocky Mountains have resulted in significant changes to short- (Marshall, 2014) and long-term hydrology (Edwards et al., 2008; Jost et al., 2012), runoff (Stewart et al., 2004), and water storage (Mote et al., 2005; Whitfield, 2001). These changes impact minerotrophic wetlands, which can be sensitive to variations in hydrology, for example, since the 1950s the montane cordilleran ecozone has experienced precipitation decreases in southern (20 ~ 50%)

and increases (30 ~ 50%) in northern regions (DeBeer et al., 2016). Changes in the phase of precipitation have also been observed over the last 60 years by Zhang et al. (2000), Schnorbus et al. (2014), and Vincent et al. (2015). On an annual basis, the authors found that the ratio of seasonal snowfall decreased by about 8% in the south and increased by ~12% in the north. The major changes occurred during spring, with reductions of ~20% for the entire region. Furthermore, snow accumulation and duration have also decreased due to a positive trend in air temperature (+1℃ since 1900s) (Zhang et al., 2000; Valeo et





al., 2007; Whitfield, 2014), which is leading to earlier and faster snow- and glacier-melt during spring, resulting in high and shortened peak flows.

By mid-century, peak flows are predicted to increase with a shift to earlier spring runoff. For example, DeBeer et al., (2021) suggest that the timing of runoff could occur up to two to four weeks earlier by 2100. Earlier snowmelt increases the length of the summer period with associated higher air temperatures and evaporative losses (Foster et al., 2016; Leppi et

al., 2012). Greater drying potential and diminished summer and autumn stream flow could have broad impacts on flora and fauna of minerotrophic montane wetlands (Stewart, 2009).

Montane floodplains and the wetlands that exist within them are governed by pulses or short intervals of water runoff, which contribute to flooding (i.e., flood pulses) (Junk et al., 1989). The flood pulse enhances biological productivity and diversity in these ecosystems (Hughes, 1997) associated with the combined effects of the flood timing, water

temperature, nutrient content, turbidity, and hydrological connectivity (Stanford et al., 2005; Lacoul and Freedman 2006; Bayley and Guimond 2008). Higher amplitude events that occur over shorter time periods or earlier in the season can inhibit the growth of some species or may initiate succession (Bayley, 1995). For example, during high flood events (wet years) Amoros and Bornette (2002) observed that fast flowing river discharge can carry away organic nutrients and deposit silt in the basins, which according to Sparks et al. (1990) and Bayley and Guimond (2009), may lead to decreases or loss of

biodiversity, marsh burial, and a change in the wetland. However, in following years, marsh can grow back as the tubers remain and can regenerate following flooding (Hernandez and Mitsch, 2006). In contrast, periodic dry periods enhance shrub growth, which can be decimated during wet periods (Takaoka and Swanson, 2008).

It is crucial to ascertain whether there is a longer-term trend in the changes that are occurring to these montane floodplains or if there are events of such a magnitude that causes this environment to move into a new ecosystem state.

Recent trajectories in montane floodplain wetland landcovers remains a source of uncertainty, which raises questions over how floodplain riparian vegetation, permanent open water, and discharge properties have increased or decreased over recent decades. Wetland land cover mapping, management and change assessments typically employ field observation and data collection (Millar et al., 2018; Ray et al., 2019; Windell and Segelquist, 1986); however, this approach is costly, labor-intensive, and unable to represent past conditions (Chasmer et al., 2020).

In this context, remote sensing (RS) data and especially the Landsat time-series, can assist in wetland trend and change analysis by providing at least four decades of data (Ju and Masek, 2016; Wulder et al., 2022). The Landsat archive, which is now longer than pulses of seasonal or interannual hydro-climatological anomalies, permits evaluation of longer-term trends across large and spatially continuous areas, to help us better understand the patterns, direction, rates and drivers of change in dynamic montane wetland ecosystems.

The Columbia River floodplain in Canada represents a unique environment to assess wetland ecosystem changes over time associated with climatic and land-use changes. Wetlands of the Columbia River Basin provide important ecosystem services, such as critical habitats for flora and fauna, such as spawning grounds of fish (Cooper et al., 2017), support food webs (Díaz et al., 2015), filter and store sediment from runoff erosion events (Lottig et al., 2013), and



accumulate and release carbon (Hrach et al., 2022). A better understanding of the trends and changes in this study region will serve as an important reference for other similar wetlands in the Rocky Mountains.

The primary goal of this study is to quantify floodplain wetland response to changing hydroclimatic conditions within the Upper Columbia River Wetlands (UCRW) in Canada during the past 39 years (1984 - 2022). The objectives are to quantify and evaluate the historical trends and changes in i) areal floodplain landcover (open water, marsh, wet meadow, and riparian shrubs and trees) extents within the UCRW, and ii) the peak flow conditions in the Columbia River over the last 39 years in terms of discharge, water level, maximum inundation extent, timing, and duration using remote sensing and river runoff observations. This study provides a framework for evaluating effects of climate change in the UCRW, supporting regional decision-makers as part of a strategic planning for the local biota and water resource management for the entire Canadian Columbia River.

## 2. Methods

### 2.1 Study area

This study focusses on a ~ 120 km stretch of the Upper Columbia River floodplain (188 km²) between Donald and Invermere within the Rocky Mountain Trench, British Columbia, Canada (Figure 1). The region drains an upstream area of ~6,660 km², presents historical averages of: air temperature 3°C, precipitation 800 mm.year$^{-1}$ (MacDonald and Chernos, 2020), wind speed ~1 m.s$^{-1}$, relative humidity 55% (Hersbach et al., 2020) and annual average peak flow of 512 m³.s$^{-1}$ (Carli and Bayley, 2015).





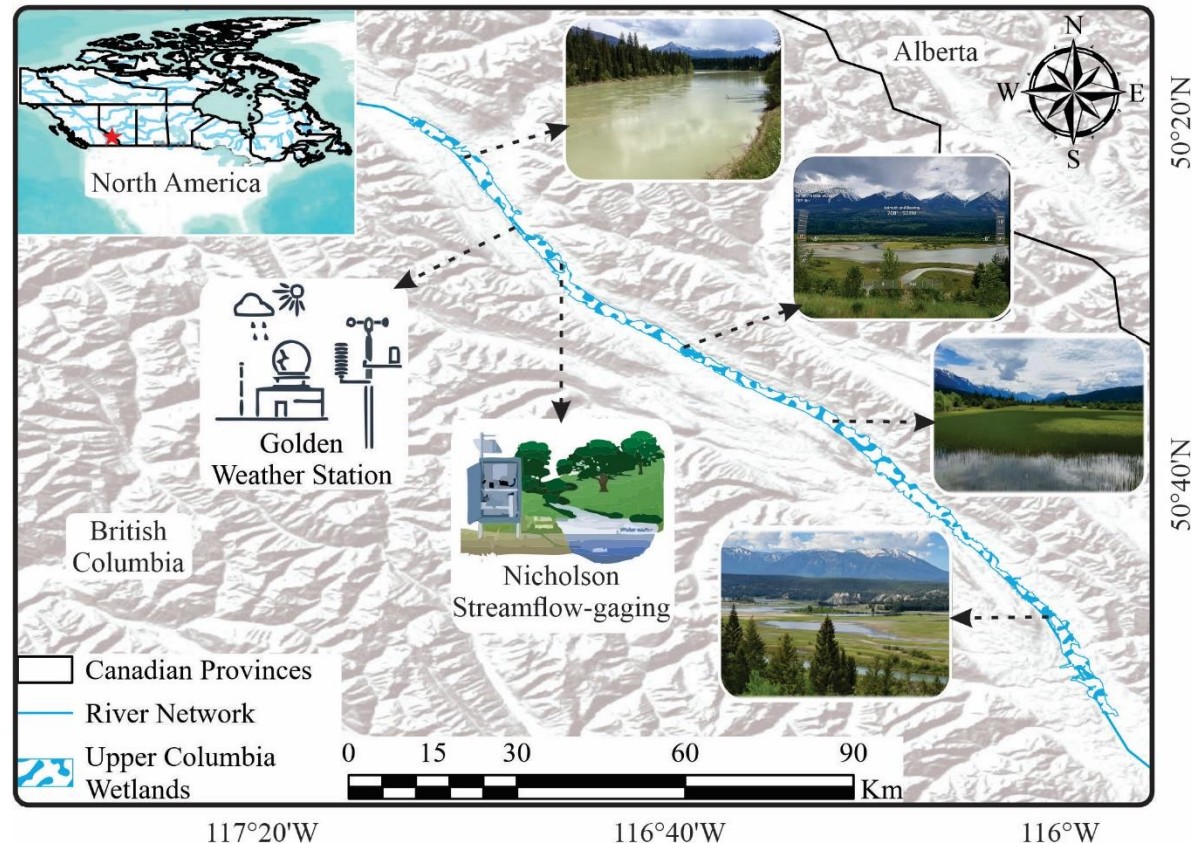

**Figure 1:** Study area and approximate location of the streamflow-gaging and weather station

## 2.2 Remote sensing and hydroclimatic data input

The investigation period was 1984 to 2022, based on the Landsat 5 (TM) and 8 (OLI) Collection 2, Tier 1, Level 2 reflectance archives available via the Google Earth Engine (GEE). GEE contains the Landsat processing methods to compute at-sensor surface reflectance, and cloud-free composites. Higher spatial resolution (e.g. Sentinel 2, European Space Agency; Airborne LiDAR - Columbia Wetlands Stewardship Partners and the Provincial Government; UAV LiDAR, and geotagged oblique photos; historical aerial photographs (BC Government, 2022), and the historical land cover classification of Canada (Hermosilla et al., 2022) were used together as a reference dataset. To determine the 'best available pixel' cloud-free pixels were selected and the median reflectance product was calculated to generate three composition images for each year: 1. prior to seasonal flooding (April to mid-May) – Spring; 2. during the peak discharge period (late-May to end July) – Summer; and 3. late summer hydroperiod (August to mid-September) – Late summer. Air temperature and precipitation data for Golden, BC (1984 - 2022) (Golden A, 1173209; Location: 51° 17' 57" N, 116° 58' 56" W) (Environment Canada, 2022a), and river discharge and water level at the Nicholson gauge (Columbia River at Nicholson, 08NA002; Location: 51° 14' 36" N, 116°



54' 46" W) (Environment Canada. 2022b) on the Columbia River (1903 - 2022) were obtained from the Environment Canada online data archives.

## 2.3 Workflow

The UCRW trend and change analysis workflow adopted seven steps, as shown in Fig. 2: i) acquisition of remote sensing data, and ii) hydroclimatic data; iii) reference dataset for classification training purposes; iv) land cover classification; v) validation of land covers; vi) trend analysis; vii) and land cover change assessment.

**Figure 2:** Methodological workflow for the spatial-temporal (1984 - 2022) analysis of vegetated and water landcover classes using remote sensing and hydro-climatological data.





### 2.4 Random Forest algorithm and training data collection

To determine changes in the vegetation- and water-extent over time, a random forest (RF) classification (Breiman, 2001) was performed using GEE. RF is a supervised machine learning algorithm that generates multiple decision trees to create and predict a raster classification, in this case four classes: Open water, Marsh (i.e., Bulrush – *Schoenoplectus tabernaemontani* ,and Cattail Marsh – *Typha latifolia*), Wet meadow (e.g., Beaked Sedge – *Carex rostrata*, Water Sedge - *Carex aquatilis*, Horsetail - *Equisetum arvense*), Woody/Shrub vegetation (e.g. Woody: Cottonwood – *Populus*, Norway Spruce – *Picea abies*, and Dogwood – *Cornus* spp.; Shrub: Sitka Willow - *Salix sitchensis*, Red – Osier Dogwood – *Cornus sericea*, Horsetail – *Equisetum* spp.). RF was used because it is non-parametric and does not rely on a priori knowledge of the ecological drivers or characteristics of the prediction/classification outputs (Menze et al., 2009).

Training data collection, however, can be challenging over large or mountainous regions, as these ecosystems have dynamic or unpredictable weather, are remote and difficult to enter (e.g. Inglada et al., 2017). Moreover, access to training data is difficult to acquire over time because data may not be available for the period of assessment; land cover classes or observations could also be different from the current study, making it difficult to compare. In this research, we utilise a variety of reference remote sensing data sources: UAV and Airborne LiDAR, aerial photographs, geotagged photos, Sentinel 2, and historical classified land cover (Hermosilla et al., 2022) to generate training samples per each year (Figure 1).

To extract or determine the most reliable training pixels within areas of unchanging landcover class, the time series classification of Hermosilla et al. (2022) was used. Land cover permanence was calculated by summing the number of times each land cover class pixel was identified in the same pixel location. Reference rasters contain a numerical pixel value (i.e. 1 – open water; 2 – marsh; 3 – wet meadow; 4 – woody/shrub) that corresponds to each land cover in the input rasters. The 1984 land cover raster was chosen as the reference raster because this was the first year of the record, thereby providing a baseline or starting point from which to compare. The permanent land cover raster was then used within GEE to mask out permanent zones within the study floodplain that showed potential as training areas. Training pixels were then allocated within these training areas and used over the whole time-series. However, in the years with available higher resolution imagery (i.e., sporadically throughout the time series: Aerial photographs – 1984 to 1991, 2005, 2007, and 2009; Sentinel 2 – 2016 to 2022; Airborne LiDAR – 2018; UAV LiDAR and geotagged photos – 2022), which by expert interpretive identification of land cover class was possible to increase the number of training pixels in these years with more reference datasets.

To reduce the uncertainty in the training data associated with classification errors in the historical land cover classification (Hermosilla et al. 2022), pixels within land cover patches were selected. Therefore, training pixels had to be at least 90m from adjacent landcovers (to reduce the potential for edge effects and mixed pixels) (Pelletier et al. 2016). The RF model was trained using 1500 trees (as per Millard and Richardson, 2015), and each class sample had a minimum of 60 pixels identified (60 pixels per land cover class; a total of 240 for the four land cover classes), and in the years with available higher resolution imagery, >40 pixels per land cover class were allocated (about 100 pixels per land cover class; a total of 400 assuming the four land cover classes), with 70% used for training and 30% reserved for validation. Pixels reserved for





training within the RF model were those that were furthest in distance from clouds or cloud shadow boundaries, as applied in White et al. (2014). The training pixels were randomly distributed across the study floodplain with each scene mosaic. Classification was performed using the five following Landsat TM and OLI bands: Blue (Band 2 in OLI; Band 1 in TM), Green (Band 3 in OLI; Band 2 in TM), Red (Band 4 in OLI; Band 3 in TM), Near Infra Red (NIR, Band 5 in OLI; Bands 4 and 5 in TM), and Short Wave Infra Red (SWIR, Bands 6 and 7 in OLI; Band 7 in TM). Figure 3 illustrates the training and validation steps and the location of the training sites.

**Figure 3:** Steps for land cover prediction: (i) Allocation of training pixels method; (ii) Accuracy assessment for the predicted land cover; (iii) Statistical evaluation using the Kappa coefficient.



## 2.5 Reference dataset and accuracy measurement

The higher spatial resolution RS, such as UAV and airborne LiDAR, aerial photographs, Sentinel 2, geotagged photos, and the non-changing pixels of the Historical Land Cover of Canada created by Hermosilla et al. (2022) (supplementary material) were used to allocate the training pixels/polygons (for more details see the supplementary material). Thereafter, the Kappa coefficient was calculated to estimate the accuracy of the random forest simulated land cover.

According to Congalton and Green (2019), 50 random ground sample points are enough for each land cover category, though, a minimum of two-hundred samples were used per mosaic image. The results of the kappa accuracy assessment were then summarized in a confusion matrix with omissions and commissions for all classes and periods. Furthermore, as discharge and open water area are expected to covary, a linear regression between these two variables was performed: i) as an additional check on the open water classification; and ii) to create a discharge-based model of floodplain inundation area.

## 2.6 Trend and change analysis

To assess the trends over 39 years in the wetland area classification and the hydroclimatic data, the Mann-Kendall (Mann, 1945, Kendall, 1975) test was performed using pyMannKendall (Hussain and Mahmud, 2019). The Mann-Kendall method is a nonparametric test used to identify a trend in a series. To evaluate changes in wetland extent, three hypotheses were tested based on trends over the period of data observation: i) no trend exists over the time period; ii) a positive trend exists; and iii) a negative trend exists. A significance level or p-value $\leq 0.05$ was assumed. For the land cover change assessment, the Change Detection Wizard in ArcGIS Pro was used with a pixel value change method to assess the shift during 1984 ~ 2022 raster datasets.

## 2.7 Discharge timing, duration, frequency, hydrograph, and anomaly assessment

To understand how hydro-climate variability might have influenced or altered wetland extents throughout the time of study, the use of direct observation (stream-gaging) methods is an appropriate way to evaluate river-based wetland changes. The timing and length of the peak flow were determined using the historical streamflow-gaging station from Nicholson (1903–2022). The data were divided using a twenty-five-year time interval to assess when the peak flow occurred and how it has changed since 1903 (compared groups: 1903 to 1928, 1929 to 1953, 1954 to 1978, 1979 to 2003, and 2004 to 2022). This time interval was chosen because the Pacific Decadal Oscillation (PDO) influences precipitation and air temperature in the central-eastern Rockies (Linsley et al., 2015), and the PDO periodicities/cycles were most energetic/perceptive within a 25 year interval average (Mantua and Hare, 2002). To determine whether the peak flow is occurring earlier or later, an average of the Julian day peak flow for each group was calculated and then compared.

The 10% highest flow (relative to peak flow) for each year was determined for the peak flow duration/length, and the number of days. To ascertain if the number of peak flow days is increasing or decreasing, the data were further divided





into the five groups. The average of the number of peak flow days per 25-year time interval was determined and then compared.

To define the distribution of river discharge and how it changed over the course of a century, a frequency (%) curve from the daily and peak discharge was built, using the same five groups (1903 to 1928, 1929 to 1953, 1954 to 1978, 1979 to 2003, and 2004 to 2022), to show the frequency of each flow discharge, and the frequency with which it is overcome. Additionally, a peak flow hydrograph analysis was carried out for the five groups using the 25-year period interval average to compare shape and how it has altered since 1903. To separate the peak- from base-flow the recession method (Brutsaert and Nieber, 1977) was used (Equation 1).

$$Q_t = Q_0 K^t \tag{1}$$

Where $Q_t$ is the baseflow (the threshold utilized was 50 $m^3.s^{-1}$ because it was observed that the discharge normally only increased beyond this value), $Q_0$ is the initial baseflow (at the beginning of the storm event, time = 0), k = exponential decay constant (ratio of the baseflow at time t = 0 to the baseflow one day earlier), and $t$ is the number of days after the start of the peak flow. This method is used to discover the daily baseflow during the peak flow, and Equation 2 is used to determine the peak flow runoff ($Q_{PR}$).

$$Q_{PR} = Q - Q_t \tag{2}$$

After finding $Q_{PR}$, the values were plotted to create a hydrograph, an exponential model (Equation 3) was generated for each hydrograph to determine the recession constant ($\alpha$) where a larger α represents a steeper decline (Berhail et al., 2012).

$$Q_a = Q_i e^{-\alpha t} \tag{3}$$

$Q_a$ is the discharge at time $t$ after recession, $Q_i$ represents the discharge at the start of recession, $e$ is Euler's number (2.71828). In terms of the anomaly evaluation was used to identify how climate change affected peak discharge. The anomaly assessment was also used to detect the predominant PDO pattern (i.e., warm, normal, or cold) in each 25 years interval in the UCRW. We employed a straightforward technique suggested by Genz and Luz (2012). The anomaly for the peak flow was determined as follows in Equation 4:

$$Anomaly = \frac{(Q_i - Q_m)}{\sigma} \tag{4}$$





$Q_i$ is the annual peak flow (m³.s⁻¹) in a year, $Q_m$ is the historical average of peak flow (m³.s⁻¹), and $\sigma$ is the standard deviation (m³.s⁻¹). The Anomaly can be classified as: Wet > 0.5; Normal year ±0.5; Dry < -0.5. The Mann-Kendall trend test

was then used to assess the anomalies to determine whether or not the peak flow anomaly was increasing or decreasing.

## 3. Results

### 3.1 Random Forest classification accuracy

A RF classification was used to determine the land cover extent per season in the UCRW since 1984 to 2022. To
do this, 100 supervised classifications were performed for each period of Spring (37), Summer (29), and Late summer (34), using a total of 32,880 pixels for all seasonal images (23,016 for training, and 9,864 for validation). The average Kappa coefficient for all land cover classes and from each study period was 0.83 (April to mid-May; Table S1 to S37), 0.85 (late-May to July; Table S38 to S66), and 0.78 (August to mid-September; Table S67 to S100), which results in an average for all images of 0.82 (all Confusion Matrix with total accuracy, omission and commission for all classes and periods, and its
corresponding classified raster are attached in the supplementary material, Table S1 to S100). Furthermore, the area of open water was correlated with river discharge for each year (Figure 4), varying from $R^2$ of 0.75 (April to mid-May and August to mid-September) to 0.82 (late-May to July). This is not a direct validation of the open water classification; however, as river discharge increases, open water extent across the floodplain also increases.

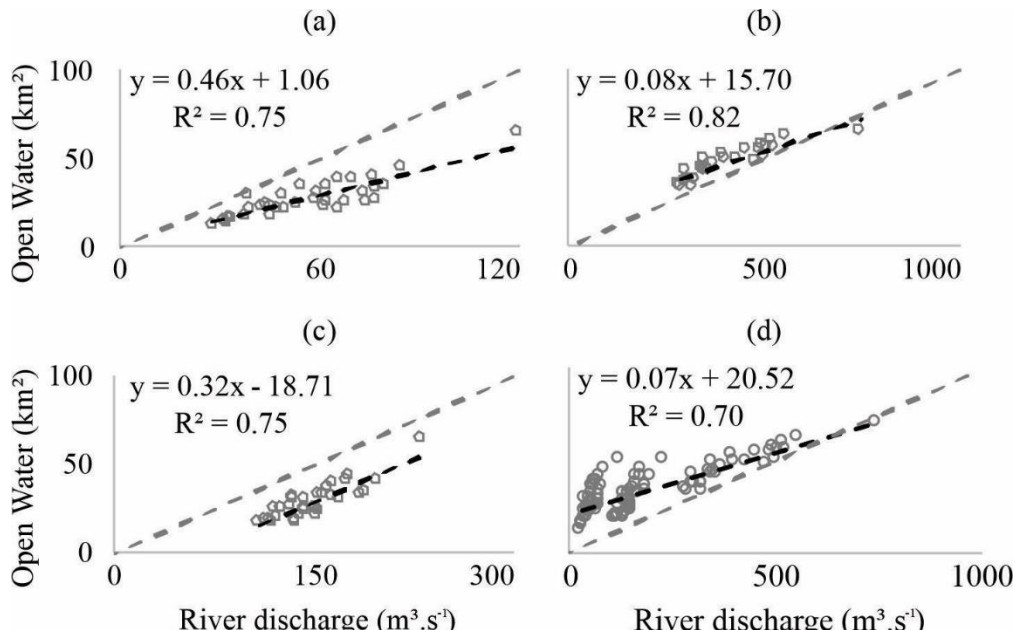

**Figure 4:** Linear regression between area of open water and river discharge in April to mid-May (a), late-May to July (b), August to mid-September (c), annual basis (d).



### 3.2 Changes in the Upper Columbia floodplain from 1984 to 2022

The area of open water decreased from April to mid-May (-0.11 km².year$^{-1}$; -5 km$^2$ or -3% in 39 years, compared to the UCRW area) and from August to mid-September (-0.28 km².year$^{-1}$; -11 km$^2$ or -6%), which might be related with an overall reduction in precipitation (-0.75 mm.year$^{-1}$; p-value = 0.01) and an increase in air temperature (0.02°C.year$^{-1}$; p-value = 0.02) since 1984. The decrease in precipitation may have resulted in a loss of marsh areas of -1.39 km².year$^{-1}$ (-55 km$^2$ or -29% during spring) and -0.24 km².year$^{-1}$ (-9 km$^2$ or -5% by late summer), and an increase of wet meadow area during spring

(1.28 km².year$^{-1}$; +48 km$^{2;}$ +26%) and reduction in the late summer (-0.12 km².year$^{-1}$;-4 km$^2$ ;-2%). In contrast, woody/shrub vegetation increased in area over the spring, peak flow period and late summer by 0.26 km².year$^{-1}$ (+11 km$^2$; +6%), by 0.50 km².year$^{-1}$ (+20 km$^2$; +11%) and 0.57 km².year$^{-1}$ (22 km$^2$ ; +12%) respectively.

     During the peak flow season (late-May to July), the open water extent showed a positive tendency (0.13 km².year$^{-1}$; +3%), likely due to the increase in peak discharge (0.58 m³.s$^{-1}$.year$^{-1}$; p-value = 0.02) and water level (0.63 cm.year$^{-1}$; p-

value = 0.02), and may be related to the increase in air temperature, which influences the beginning of the snowmelt period. This rapid and earlier rise in open water during the summer may have a negative impact on marsh growth which may explain the negative trend of -0.90 km².year$^{-1}$ in marsh area in this period (-19%) in the floodbasin. The marsh areas declined and were replaced by more open water (0.13 km².year-1; +3%), and an increase in the area of wet meadow (0.30 km$^2$.year-1; +7%) and woody/shrub (0.50 km$^2$.year-1; +11%) over the 39 years in the summer period.

The trends of the land cover extent and hydroclimatological indicators are shown in Figure 5 and Table 1, respectively. The overall annual trends show that open water (-0.03 km².year$^{-1}$) and marsh (-0.10 km².year$^{-1}$) extent are decreasing, whereas wet meadow (0.06 km².year$^{-1}$) and woody/shrub (0.17 km².year$^{-1}$) areas are expanding (Figure 5d). The relatively small annual changes compared to the larger seasonal changes (Figure 5 d vs 5 a,b,c) demonstrates the importance of the seasonal evaluation in detecting changes in the UCRW. The spatial context and the percentage of change during

spring, summer, and late summer in Figures 6, 7, and 8, shows how each land cover has changed over 39 years in the floodplain.. The location/coordinates of the main changes are provided in the supplementary materials file (Tables S101 to S104).





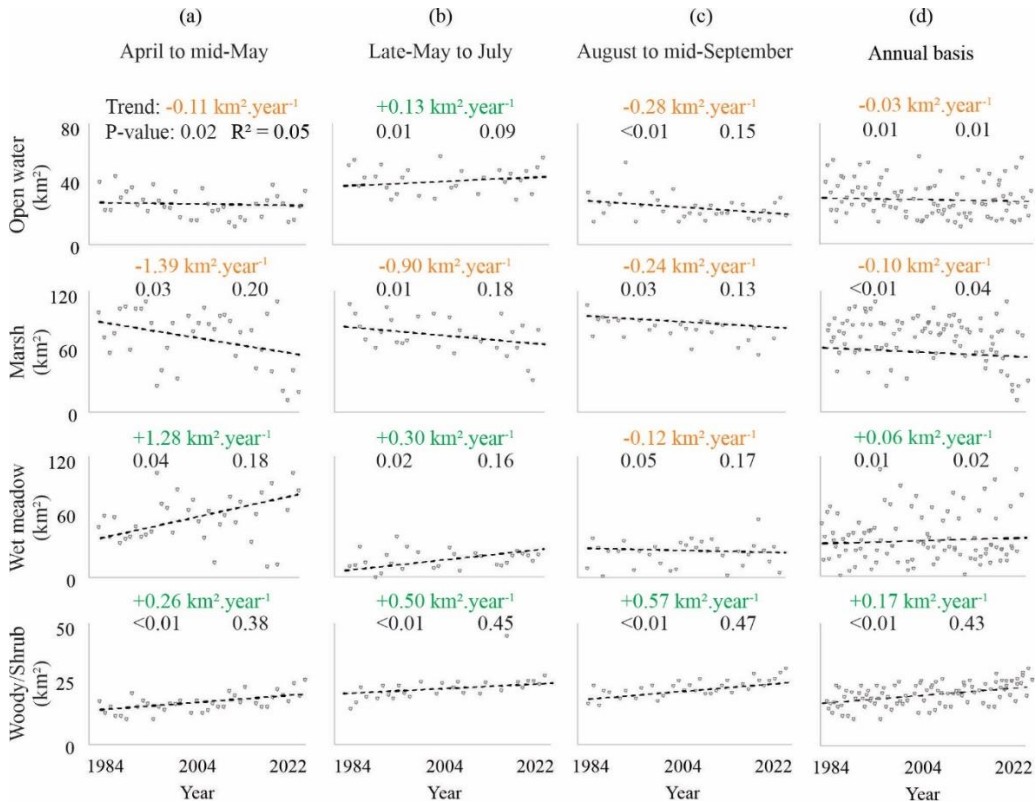

Green – Positive trends; Orange – Negative trends.

**Figure 5:** Trends of land cover extent during April to mid-May (a), late-May to July (b), August to mid-September (c) and in on annual basis (d) (1984 – 2022)

**Table 1.** Historical and trends of hydroclimate variables (1984 - 2022).

| Variable | Air temperature | Precipitation | Discharge | | Water level | |
|---|---|---|---|---|---|---|
| | | | Annual | Peak flow | Annual | Peak level |
| Min. | -30.10 °C | 301.40 mm | 12.70 $m^3.s^{-1}$ | 283 $m^3.s^{-1}$ | 10.10 cm | 232.20 cm |
| Mean | 5.22 °C | 463.80 mm | 104.97 $m^3.s^{-1}$ | 428.18 $m^3.s^{-1}$ | 106.20 cm | 320.70 cm |
| Max. | 25.70 °C | 641.40 mm | 748 $m^3.s^{-1}$ | 748 $m^3.s^{-1}$ | 421.40 cm | 421.40 cm |
| $\tau$ | 0.02 | -0.08 | 0.02 | 0.04 | 0.06 | 0.07 |
| S | 1197480 | -62 | 1382135 | 25 | 5098418 | 45 |
| p-value | 0.02* | 0.01* | <0.01* | 0.02* | <0.01* | 0.02* |
| Slope | 0.02 °C.year$^{-1}$ | -0.75 mm.year$^{-1}$ | 0.01 $m^3.s^{-1}$ year$^{-1}$ | 0.58 $m^3.s^{-1}$ year$^{-1}$ | 0.01 cm.year$^{-1}$ | 0.63 cm.year$^{-1}$ |

Min. – Minimum; Max. – Maximum; * – There is a significant temporal trend at the 5% level; S and $\tau$ – indicate the trend (negative or positive); Slope – represents the 39 years increase/decrease of the variable; p-value – trend significance ≤ 0.05 high significance.



**Figure 6:** The Upper Columbia River floodplain distribution of land cover change from April to mid-May. Map insets (a) indicate changes in land cover for sample regions. Sankey diagram (b) of the changes in land cover from 1984 to 2004 (left) and 2004 to 2022 (right), the land cover change since 1984, and the percentage of change compared to the Columbia wetlands area (188 km²).


**Figure 7:** The Upper Columbia River floodplain distribution of land cover change from late-May to July. Map insets (a) indicate changes in land cover for sample regions. Sankey diagram (b) of the changes in land cover from 1984 to 2004 (left) and 2004 to 2022 (right), the land cover change since 1984, and the percentage of change compared to the Columbia wetlands area (188 km²).







**Figure 8:** The Upper Columbia River floodplain distribution of land cover change from August to mid-September. Map insets (a) indicate changes in land cover for sample regions. Sankey diagram (b) of the changes in land cover from 1984 to 2004 (left) and 2004 to 2022 (right), the land cover change since 1984, and the percentage of change compared to the Columbia wetlands area (188 km$^2$).



### 3.3 Upper Columbia River discharge

The average discharge ($Q_m$) of the Upper Columbia River was 428.2 m³.s⁻¹, with standard deviation ($\sigma$) of 105.9 m³.s⁻¹ and these values were used to classify the peak discharge in the anomaly assessment. The twenty-five-year interval

represented well the cold and warm Pacific Decadal Oscillation (PDO) pattern at the UCRW, and how this atmospheric teleconnection affected local river discharge. Figure 9 shows the anomaly time series of streamflow values in the Upper Columbia River. Classification of the annual events by the anomaly method resulted in forty-one normal years, thirty-seven wet years, and forty-one dry years. Additionally, analysis of the anomaly values with the Mann-Kendall test yields a negative trend of -0.08 (p-value = 0.02; $\tau$ = -0.03; S = -209), showing a dry tendency, which is consistent with the decline of

open water area and increase in woody/shrub vegetation during the 1984 to 2022 period.

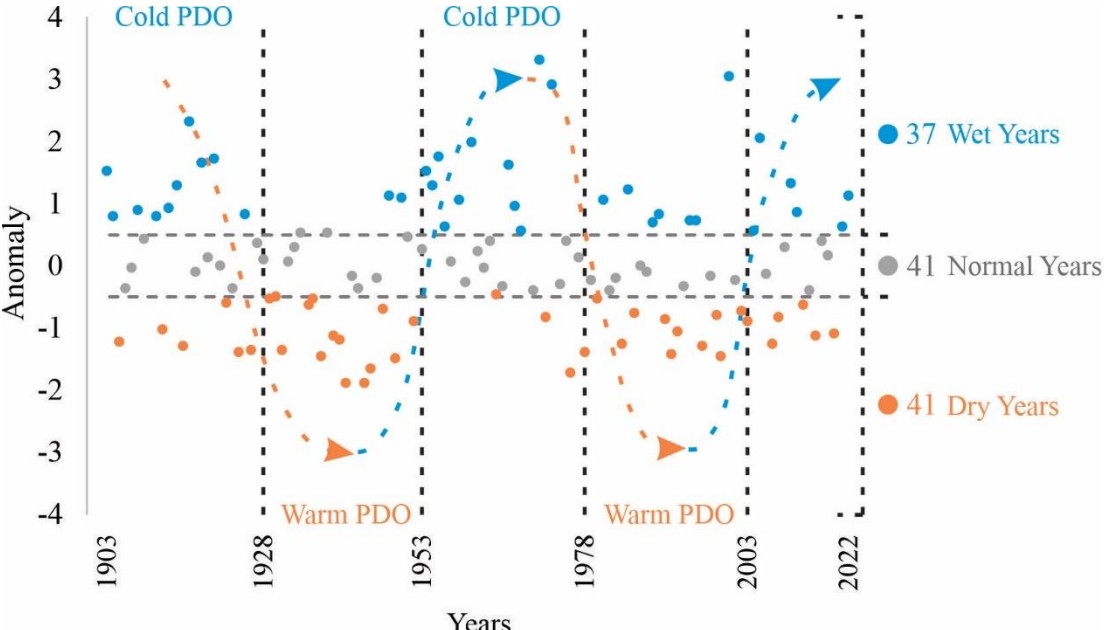

**Figure 9:** Anomaly time series of annual peak flow upstream of the Upper Columbia River, British Columbia, Canada, and the predominant PDO phase in each 25 years

The timing of the peak flow ( Julian peak flow days) may be explained by the predominant PDO phase since 1903 (Table 2). Peaks flows tended to be late in the Cold PDO (June 26, June 22, and June 15; as consequence of colder air temperatures) and earlier during the Warm PDO (June 20, and June 16; due to higher air temperatures). However overall, regardless of the PDO phase, the Julian peak flow day is starting earlier in the season (Table 2).

Table 2 summarizes the peak flow day for the Upper Columbia River and its duration since 1903. From 1903 to

1928, June 26 (Julian Day 178) was found to be the approximate day of annual peak flow. From 2004 to 2022, peak flow occurred on average on June 15 (Julian Day 169), eleven days earlier than in the past (i.e., 1903 to 1928). Peak flow duration also changed from an average of 22 days from 1903 - 1928 to 11 days after 2003. In contrast, if 1979 to 2003 is compared



with 2004 to 2022 (the period of our remote sensing dataset), the peak flow was only earlier by one day, and its duration was shortened by one day. Thus, in the last century, the peak flows have gotten earlier in the season and the duration of peak flow
shorter resulting in a dryer floodplain during the summer growing season.

**Table 2.** Annual average peak flow day for each PDO group (1903 to 1928, 1929 to 1953, 1954 to 1978, 1979 to 2003, and 2004 to 2022) and the duration/length of each peak flow period.

|  | Julian peak flow day (date) | Var. | SD | Duration/Length of peak flow period (days) | Var. | SD |
|---|---|---|---|---|---|---|
| 1903 - 1928 | 178 (June 26) | 218 | 15 | 22 | 115 | 11 |
| 1929 - 1953 | 172 (June 20) | 224 | 15 | 15 | 80 | 9 |
| 1954 - 1978 | 174 (June 22) | 177 | 13 | 12 | 46 | 7 |
| 1979 - 2003 | 168 (June 16) | 166 | 13 | 12 | 56 | 8 |
| 2004 - 2022 | 167 (June 15) | 133 | 11 | 11 | 51 | 7 |

Var. – Variance; SD – Standard deviation

The frequency analysis revealed that higher peaks appeared more frequently, since the lowest peak from 1903 to 1928 was 280 m³.s⁻¹, while the lowest peak after 2003 was 305 m³.s⁻¹, which was 9% higher (Figure 10a). Moreover, the daily discharge frequency (Figure 10b) revealed that 1903 – 1928 period had a higher discharge rate than post-2003 in terms of frequencies between 10% and 60%.

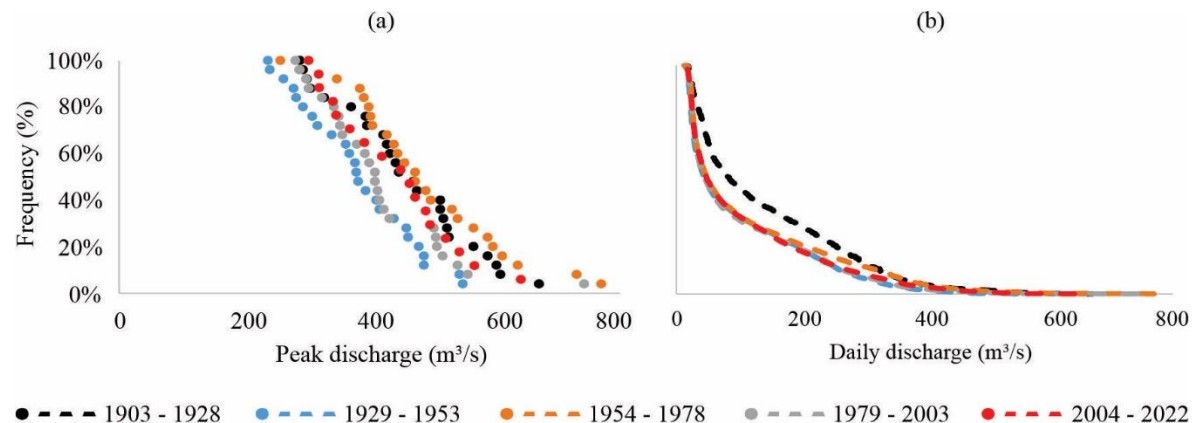

**Figure 10:** Frequency assessment of the Upper Columbia River peak (a) and daily discharge (b) from 1903 to 2022

Figure 11a shows that the peak flow hydrograph had a broader shape during the intervals of 1903 to 1928 and 1929 to 1953, with a lower discharge, before becoming steeper with larger flows in a shorter amount of time. The positive trend in the recession constant coefficient over time, which is depicted in Figure 11b, can also explain this pattern.





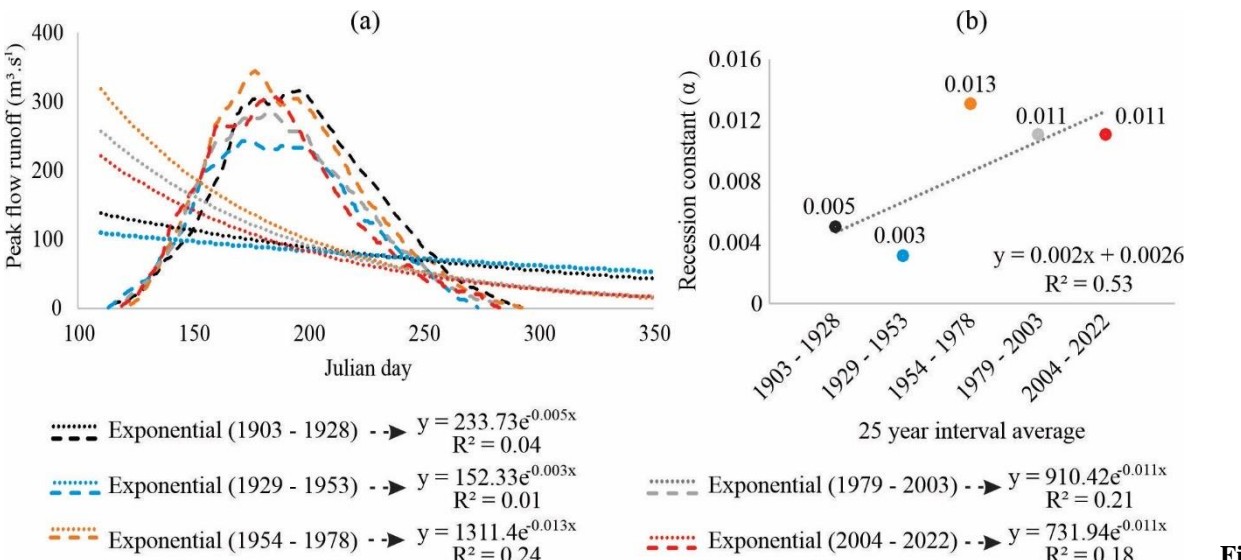

**Figure 11:** Twenty-five-year interval average peak flow runoff hydrograph of the Upper Columbia River (a), and hydrograph recession constant tendency (b) (1903 to 1928, 1929 to 1953, 1954 to 1978, 1979 to 2003, and 2004 to 2022)

## 4. Discussion

### 4.1 Classification evaluation

The land cover classification used in this study was made possible by the utilisation of various types of data for calibration and validation, resulting in the annual kappa average of 82%, with higher precision during April to mid-May and the peak discharge period. At the beginning of the growing season (April to mid-May) the four classes (Open water, Marsh, Wet meadow, and Woody/Shrub) were easier to distinguish. During the peak flow period, the floodplain surface is mostly covered by water, marsh, and woody vegetation, with a small area of wet meadow visible since the water levels were often so high that the dominant wetland meadow vegetation was covered. During the late summer (August to mid-September) there is higher greening in the Marsh and Woody/Shrub vegetation, which creates some confusion between the vegetated classes. Moreover, marsh and wet meadow merge, which might be contributing for the lower kappa during April to mid-May and August to mid-September.

In addition, based on the high correlation between river discharge and open water in the Upper Columbia River floodplain, the open water area can be estimated from river discharge. This result is consistent with Hopkinson et al., (2020), which also found similar relationship over a smaller part of the UCRW, $R^2$ 0.87. However, the moderate correlation ($R^2$ 0.70) may be explained as the floodplain contains hundreds of wetlands that flood during peak flow and retain water during late summer and spring. There is substantial variation spatially and temporally in flood depth because some years the flood peak overtops the natural levees into all the wetlands while in other years, water reaches into the wetlands through natural channels or gaps in the natural levees. This results in a highly dynamic hydroperiod that influences vegetation communities in a wide range of ways.





Each year in early spring, the wetland water levels are at their lowest levels, when the Columbia River drops to 0.3 meters (Environment Canada. 2022b), and according to our results there is a negative trend of open water during this season. If the open water areas keep shrinking, marsh and wet meadows would be expected to reduce their area as well, and woody/shrub vegetation would be expected to encroach in the floodplain, which is what has been observed in all seasons. In contrast, when the flood pulse rises 2 – 4 meters, much of the wet meadow vegetation is covered by the turbid floodwaters.

The UCRW has connected and non-connected wetlands, with both types experiencing variation in water levels and areal extent as the river rises and falls; however, more isolated wetland water bodies may suffer permanent level and extent changes as a result of overbank flooding, slow drainage, loss to evapotranspiration (MacDonald and Chernos, 2020; Carli and Bayley, 2015), and climate change (Bürger et al., 2011; Carver, 2017; Jost et al., 2012). The aforementioned effects have primarily been seen between April and mid-May, and August to mid-September, when open water is shrinking, and woody/shrub vegetation is encroaching within the floodplain.

## 4.2 Hydrometeorological changes in the upper Columbia River Basin

The positive trend of air temperature in the UCRW (0.02 ℃/year) is consistent with an increase in global air temperatures during the same period (Hansen et al., 2010; Ohmura and Wild, 2002). Between 1900 and 1998, Western Canada warmed by ~ 1 ℃ (Zhang et al., 2000), and since the early 1960s, the trend on the eastern slopes of the Canadian Rockies has been warmer than the regional norm (2.6 – 3.6 ℃) (Harder et al., 2015). The proportion of rainfall to total precipitation is predicted to increase as air temperatures increase while the proportion of precipitation that falls as snow tends to decrease (Lapp et al., 2005). For the Canadian Rockies, trends in precipitation are mixed, with some studies reporting increasing trends of roughly 14% during the period 1948–2012 (Vincent et al., 2015) and other studies finding neither trends nor change (Valeo et al., 2007), as well as a declining trend (-0.75 mm.year$^{-1}$) as was observed in this study.

Streamflow is the basin-scale integrated response to these hydrometeorological changes, and in high elevation headwater regions with limited meteorological monitoring, streamflow is a readily observable indicator that can be used to gauge hydroclimatic change (Harder et al., 2015). In the past century, several natural annual stream flows in the southern Canadian Rockies have decreased (Rood et al., 2005; Brahney et al 2017), and peak streamflow events have been observed in some rivers to arrive earlier and with less flow volume, with late summer flows dropping and winter flows rising (Rood et al., 2008). The positive trend in the annual basin discharge and peak flow was also seen in the UCRW, indicating that the peak flow had shifted to eleven days earlier and that the peak flow duration had decreased by eleven days (compared with pre-1928).

Although there is a positive trend in the peak discharge, according to the frequency analysis the highest peak (770 m³.s$^{-1}$) was observed during 1903 – 1978. The lower peak discharges during post-1978 (compared with pre-1978) may be a result of the negative precipitation trend from 1984 to 2022. Furthermore, a negative trend was observed when it comes to daily discharge, mostly in frequencies between 10% and 60% which refers to the start and end of the peak discharge, respectively. According to the historical analysis of the peak flow runoff hydrograph, the reduction of these flows (between



10% and 60%) typically leads to faster and higher peak discharge over fewer days, which is consistent with the positive tendency of the recession constant value, and in accordance with Brahney et al. (2017), who in the same area observed an 11% decline in river flow between 1947 and 2011, predominantly in late summer.

Another explanation for this pattern is through the anomaly assessment, which showed that dry or warm PDO phases are becoming drier, essentially post-1978, as per Newton et al. (2014), who since the 1970s observed a more severe dry PDO phase over the Canadian Cordillera. This suggests that the peak flow is shortening, while the magnitude of the discharge is increasing, causing higher discharges over fewer days, and it may impact the water availability through the year (specifically over the late summer), which relates to the downward annual basis open water area trend. These results agree

with Hopkinson et al. (2020), who used Landsat data to calculate water extent and hydroperiod change from 1984 to 2019 in a portion of the Canadian Columbia wetlands. They found a reduction of the permanent water area extent by ~16% (or ~3.5% of the floodplain), which is higher than the decrease in open water found in this research (-6%) for the entire Columbia River valley over the year. Those aforementioned results are in accordance with other snow-driven montane ecosystems in western Canada (Burn, 1994; Burn et al., 2004; Cutforth et al., 1999; Whitfield, 2001; Barnett et al., 2005;

Stewart et al., 2005).

Under current and future climate change, earlier and faster snowmelt is expected, directly affecting the snow accumulation- and melt-dominated watersheds (Steger et al., 2013; Pörtner et al., 2019). The positive trend of the air temperature is probably the cause for the earlier snowpack melt, which increases peak flow volumes, while shortening the duration (DeBeer et al., 2021). The rapid rise of the peak flow will enhance the open water area during spring and summer,

which when combined with positive air temperature trends may increase the evaporative demand, which will impact the amount of water that is lost to the atmosphere and the whether or not these wetlands will shift into other land cover types, which can further enhance evapotranspiration (e.g. shrubs), essentially in late summer (Kienzle, 2006; Rasouli et al., 2022). The reduction of marsh and open water, as well as the increase in wet meadow and woody/shrub vegetation from August to mid-September, are also reflected in this pattern.

## 4.3 Hydroclimatic trends as drivers of land cover change

Hydrological changes may be a key factor in this trend of UCRW expansion of woody/shrub cover. Changes to the late winter flow regime will affect ice formation and break-up, a fluvial geomorphic process that creates sites for seedling colonisation by the riparian vegetation, encouraging clonal suckering (Rood et al., 2008). The woody/shrub vegetation is

increasing as the land is drier for longer and thus the floodplain water table is lower, which leads to a drier root zone, which allows the spread of woody vegetation (Liu et al., 2022; Pellerin et al., 2016).

Regarding trends of woody encroachment, Barger et al. (2011) found positive trends of 0.8% cover.year[-1] in the Northern US Rocky Mountains over a 30-year period. In the central region of the Rock Mountains of Alberta, Glines (2012) observed a positive encroachment trend of 0.9% cover.year[-1] since 1952 to 2003. In Niwot Ridge, south of the Rocky

Mountains, Formica et al. (2014) reported a positive woody encroachment trend of 0.2% cover.year[-1] over 62 years, which



was the same rate found by Tape et al., (2006) in Northern Alaska. The average annual positive trend (0.17% km.year$^{-1}$) of woody/shrub encroachment in the our UCRW study is in accordance with other studies. Moreover, the woody ecotone advance has the potential to interfere with almost all regional components of the hydrological cycle: higher evapotranspiration by woodlands (Donohue et al., 2007); increase of the rainwater interception by the canopy trees (Owens et al., 2006); lower runoff (Bonan, 2008); decreases of the groundwater recharge, streamflow (recharge below the rhizosphere) (Tennesen, 2008). The progression of wetland communities from herbaceous to woody plants is considered a natural succession (Vogl, 1969; Mitsch and Gosselink, 2000), although, climate change has accelerated the woody ecotone encroachment in some mountain wetlands (Politti, et al., 2014).

In addition to the possible effects of climate change, atmospheric teleconnection influences like El Niño and La Niña may significantly alter streamflow and impact land cover as has been found in other studies across western Canada (Gobena and Gan, 2006; Jacques et al., 2012; Fleming and Sauchyn, 2013; Chasmer and Hopkinson, 2017). El Niño appears to impact the UCRW as the frequency of this phenomenon has increased since 1980s (Cai et al., 2021; Zhou et al., 2014), which may explain the predominantly dry pattern in this region (Yang Yang et al., 2021) (number of years by anomaly: thirteen normal years, thirteen wet years, and sixteen dry years), justifying the downward trend for open water on an annual basis.

The historical air temperature and precipitation record of the UCRW, resampled for El Niño and La Niña occurrences, can be used to overlay climate projections from general circulation models, which may result in more pronounced future changes to the region's precipitation and temperature. According to the projections of the Special Report on Emissions Scenarios (Byers et al., 2022; IPCC, 2007) climate scenarios for the 2050s, the average annual precipitation of the Canadian Rocky Mountains could further decrease by about 5% while the average annual temperature could marginally increase by about 0.3ºC under the potential combined impact of both climate change and El Niño. In contrast, based on the predictions of the Special Report on Emissions Scenarios of 2050s (Byers et al., 2022; IPCC, 2007), La Niña years might see an additional 9% increase in average precipitation while a 0.3ºC decrease in average temperature. The drying effect of climate change on the UCRW should be partly mitigated by a future La Niña year, but that effect could worsen in an El Niño year. These findings support those of Gobena and Gan (2006), who found that, in south-western Canada, El Niño and La Niña occurrences, respectively, cause large negative and positive streamflow anomalies.

In western Canada, the Pacific North America (PNA) pattern is the active atmospheric teleconnection that has the greatest impact on the local climate and hydrology (Newton et al., 2014). The PNA pattern has a positive (i.e., relates to the Pacific warm episodes, El Niño, and characterized by above-average air temperatures), and a negative (i.e., associated with Pacific cold events, La Niña, marked by low-average air temperatures) phase (Lopez and Kirtman, 2019). The increased frequency of the positive phase of PNA has been noticed over the years (Gan et al., 2023; Wang et al., 2017), and the main reason might be the continuous emissions of greenhouse warming (Cai et al., 2014). A more frequent positive phase of PNA may lead to higher air temperatures, which leads to reductions in the seasonal snowpack (Mote et al., 2005; Brown and Robinson, 2011), and earlier spring runoff (Stewart et al., 2004).



Overall, the dominant changing seasonal hydrological processes within the UCRW include: shortening of snowmelt period due to increasing air temperatures, which will boost the river discharge as well as the groundwater (essentially during summer season); increasing and shortening of peak flows in summer due to shortening period of snowmelt combined with higher intensity rainfall and greater wetland flooding (Figure 12b) (Musselman et al., 2018), which may also cause increased erosion in the uplands and siltation of the floodplain (Zhang et al., 2022); by late summer, water

has moved through wetlands, causing them to dry, resulting in shrubification and enhanced water losses from evapotranspiration (Li et al., 2020) (Figure 12c).





**Figure 12:** Updated conceptual understanding of the hydrological processes conditions for the UCRW during Spring (a),
Summer (b), and late Summer (c).





The UCRW has changed over the past 39 years as a result of the rise in air temperature and decrease in precipitation, which has caused significant changes in the floodplain. Remote sensing was used in this work to identify areas with low, moderate, and large shifts since 1984 and to evaluate the spatial distribution of land cover trends and changes. The results obtained illustrate the potential for the fusion of remote sensing and hydroclimatological data for the assessment changing montane wetland environments.

## 5. Sources of uncertainty

Although the Random Forest algorithm demonstrated acceptable accuracy, several sources of uncertainty may be present regarding the training and validation data sources, and the spatial resolution of the Landsat imagery. It would be ideal to train and verify landcover classes using historical measured in-situ field data. However, due to the constraints over temporal data availability, different remote sensing data sources were necessary for model calibration and validation in this historical LC classification. The additional training pixels allocated using the higher resolution dataset are in the beginning (1984 to 1991), part of middle (2005, 2007, and 2009), and in the late of the time series (2016 to 2022), which tends to reduce the inaccuracies caused by the less accurate data in other years (i.e., unchanging pixels). Sixty unchanging pixels per LC class from the historical land cover classification of Canada (Hermosilla et al., 2022) (overall accuracy of 80%) were used as reference dataset for all years, therefore, any errors here could be propagated into the UCRW LC classification. However, the distribution of most reliable training and validation pixels (using higher resolution datasets) over time has enabled accurate results, which when compared to independent data sources, confirm the encroachment of woody/shrub vegetation (Politti, et al., 2014; Primack, 2000; Theurillat and Guisan, 2001), and reduction of open water during late summer (Rood et al., 2005; Kienzle, 2006; Rood et al., 2008) in montane wetlands.

Regarding the spatial resolution of the Landsat time series, thirty metres is not the ideal resolution for classifying wetlands because this habitat is typically mixed, with variable width ecotones and vegetation inside open water and vice versa. In this study, the use of complementary remote sensing data sources has enabled temporal calibration and validation but further enhancements may be possible by the addition of new data sources. For example, combining the multispectral sensors with synthetic aperture radar (SAR) may increase the accuracy up to 85% (Loosvelt et al., 2012; Mahdianpari et al., 2017; Muro et al., 2020), since SAR is an effective tool to identify permanent open water (Montgomery et al., 2018).

## 6. Conclusion

In this study we analysed temporal trends and changes of land-cover in the Upper Columbia River Wetlands using remote sensing and a Random Forest classification routine for a 39 year-period. The classifier delivered a reasonable level of accuracy (Kappa 82%). 39 years of rising air temperature resulted in an increase in the Columbia River discharge. During the peak flow period, open water extent showed a positive tendency (0.10 $km^2.year^{-1}$), while on an annual basis open water area is declining (-0.03 $km^2.year^{-1}$). Furthermore, the peak flow occurs one day earlier now than ~40 years ago, while peak



flow duration has decreased by one day. However, since 2003 , the peak flow has occurred eleven days earlier than 1903 – 1928, and its duration has reduced by eleven days, which has resulted in higher discharges in a shorter time. It also means that the summer period is drier and the land cover vegetation subject to drier conditions. According to the anomaly assessment approach, dry years have been increasingly frequent since 1984.

Open water areas of the floodplain have decreased in size during the April to mid-May period, while a large area
of floodplain marsh has been replaced by wet meadow. In the same period, shrub and woody vegetation have increased during the 39 years by 11 km$^2$. The peak flow period shows a decline in marsh regions and an increase in wet meadow, woody/shrub, and open water, the latter of which revealed a moderate increase. From August to mid-September, there was a decline in the amount of open water, marsh, and wet meadow, but a significant increase in the amount of woody/shrub species.

Overall, the future of the Upper Columbia River Wetlands and their ecohydrological services are at risk due to the altered runoff regime that favors drying of the floodplain. Expansion of riparian shrub and treed ecotones are gradually replacing marsh and wet meadow landcovers commensurate with a reduction in permanent open water area, which may lead to higher evapotranspiration, mainly during late summer, thus raising the potential for drought. While new floodplain riparian ecosystem habitats are being created, these come at the expense of lost open water/aquatic habitat.

**Competing interests**

The contact author has declared that none of the authors has any competing interests

**Acknowledgements**

The authors acknowledge funding provided by the Natural Sciences and Engineering Research Council of Canada (NSERC) (2017-04362), Alberta Innovates and The Columbia Wetlands Stewardship Partners.

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
