# Peer review of "Multi-decadal Floodplain Classification and Trend Analysis in the Upper Columbia River Valley, British Columbia"

_Hydrology and Earth System Sciences, 2023_

## Author Response (AR1)

**Hydrology and Earth System Science** – HESS – 2023-211

**Manuscript title:** Multi-decadal Floodplain Classification and Trend Analysis in the Upper Columbia River Valley, British Columbia

**Authors:** Italo Sampaio Rodrigues, Christopher Hopkinson, Laura Chasmer, Ryan J. MacDonald, Suzanne E. Bayley, Brian Brisco

**RESPONSE TO REVIEWER # 1 and # 2 AND EDITOR**

Dear Dra. Patrica Saco,

Editor of the Hydrology and Earth System Science,

Thank you for considering our manuscript entitled "Multi-decadal Floodplain Classification and Trend Analysis in the Upper Columbia River Valley, British Columbia" as suitable for publication in the Hydrology and Earth System Science after minor revisions.

We have revised our work according to the suggestions from the reviewer, to whom we are very grateful for the valuable comments. Please find our replies to each of the queries in the following pages, as well as the attached new version of the manuscript with all changes colour-highlighted (in yellow).

With our kind regards,

Italo Rodrigues

Christopher Hopkinson

Laura Chasmer

Ryan J. MacDonald

Suzanne E. Bayley

Brian Brisco

**Reviewer #1:** This work is focused on changes in floodplain wetlands and hydro-climatological variables over a relatively small area (188 km$^2$) in the Upper Columbia River Valley, over the period 1984-2022. The analysis mainly relies on the temporal and spatial variability of land cover observation, and the main results show a reduction of wetlands areas with increasing in shrub and woody areas over the last 40 years. I found the paper well written and well organized. I think it could be a significant scientific contribution for understanding the hydrological changes of this area. Furthermore, I appreciate the analysis on a seasonal scale, which provides detailed information about changes over different periods.

Authors: Dear Reviewer #01, we would like to thank you for the positive feedback and comments, as well as the time spent to help us improve our manuscript. Please find point-by-point answers below.

However, I think that it is very dense of information, and, personally, I mostly struggled in following the key message and results. Please consider abbreviating or shortening some part of the text, in order to provide a clear scientific message. For instance, I would condense lines 22-28 in one sentence, providing only the key message.

Authors: As suggested, we have made some adjustments to the text in the places noted below, in the hopes that this clarifies the message, as requested. In particular, we have shortened lines 22 – 28, providing the key message in the following updated statement:

> *Lines 23 – 25: "Thus, woody shrubs (+6% to +12%) have expanded as they gradually replaced marsh and wet meadow landcovers with a reduction in open water area. This suggests that increasing temperatures have already impacted regional hydrology, wetland hydroperiod and floodplain land cover in the Upper Columbia Valley."*

**Introduction**

I would include at the end of the Introduction Section a brief description of the sections of the paper. In this way it would be easier for the reader to follow the work.

Authors: As recommended, we have included a brief description of the sections of the manuscript at the end of the introduction, in lines 96 – 97.

> *Lines 96 – 97: "To achieve this, seasonal land cover classifications from Landsat 5 and 8 were generated using a Random Forest algorithm, and local hydroclimatic data were examined to help explain the observed landcover trends."*

**Section 2.7**

Why do you analyze the peak flow separately from the base flow?

Authors: We thank the reviewer for this relevant question. Peak flow was analyzed separately from base flow because it is during peak flow when isolated floodplain wetlands above natural levees tend to become inundated. These isolated wetlands are therefore more sensitive to dry periods relative to those that are connected to the river system. The aim was to demonstrate the temporal variation of the peak flow, which has tended towards earlier occurrence and shorter duration, thus impacting the open water area, and inhibiting the growth of some species in some locations or initiating succession processes in others (as per Bayley, 1995). We have updated Figure 11 which now shows that the twenty-five-year interval average baseflow discharge hydrograph of the Upper Columbia River has not changed considerably since 1903. In addition, we rephrased a sentence in the Results (section 3.3, Lines 353 to 355) and Discussion (section 4.2, Line 415), adding clear and quantifiable results about the baseflow. We believe that with this addition, it should be clear that the changes in peak flow conditions are of primary interest here.

> *Lines 353 – 355: "Figure 11a shows that the baseflow had a decrease (average of -1.5%) from October to March since 1903, and the peak flow hydrograph had a broader shape during the intervals of 1903 to 1928 and 1929 to 1953, with a lower discharge, before becoming steeper with larger flows in a shorter amount of time."*

> *Lines 414 – 416: "Furthermore, a negative trend was observed when it comes to daily discharge (which relates to the decrease in the baseflow since 1903), mostly in frequencies between 10% and 60% which refers to the start and end of the peak discharge, respectively."*

Bayley, P. B.: Understanding large river: floodplain ecosystems. BioScience, 45(3), 153-158, https://doi.org/10.2307/1312554, 1995.

And then, why do you look at the hydrograph (Fig. 11) without considering the base-flow contribution?

Authors: We thank the reviewer for this question. In Figure 11, we aimed to show only the peak flow portion, as we had already ascertained that changes to baseflow were minor by comparison and did not directly impact the floodplain inundation processes which controlled the wetland hydroperiod. However, we are grateful for the reviewer's question, as it illustrates that this was not entirely clear in our manuscript and that the baseflow condition and change should be presented. As noted above, we have updated Figure 11 by adding the baseflow discharge (threshold utilized was 50 $m^3.s^{-1}$ because it was observed that the discharge normally only increased beyond this value; lines 224 and 225). The new Figure 11 is below, and in line 357.

[Figure]

**Figure 11**: Twenty-five-year interval average river discharge hydrograph of the Upper Columbia River (a), and hydrograph recession constant tendency from the peak flow runoff (b) (1903 to 1928, 1929 to 1953, 1954 to 1978, 1979 to 2003, and 2004 to 2022)

**Section 3.2**

(line 267) Here, you assume that the reduction in precipitation, as observed locally at the gauge station, is a possible cause of the decrease in open water. I believe that the hydrological processes of the area can be significantly influenced by the upstream area (6,660 km$^2$), and local rainfall measurements may not represent significant drivers of these changes. Therefore, I would suggest relating these changes solely to discharge observations, which involve all the processes of the upstream area. What do you think about it?

Authors: We thank the reviewer for this relevant comment and question. River discharge plays a crucial role in maintaining the connections between open water areas in the Columbia wetlands. However, local precipitation (seasonal rain or spring snow melt) over the floodplain and adjacent slopes is a crucial source of water input for the more isolated or elevated open water areas across the valley (discussed in Lines 388 to 393). However, due to the increase of the air temperature and negative trends in precipitation since 1984, these lower amounts of precipitation (floodplain snowmelt in particular) are occurring earlier and over a shorter period, which directly impacts the isolated open water areas. Thus, although there is a positive trend in the annual river discharge (which is influenced by the upstream area; 6,660 km$^2$), this pattern is not followed by the open water areas, as this landcover is experiencing a negative annual trend, essentially during spring and late summer (Lines 266 to 269). Furthermore, the precipitation pattern in the Columbia wetlands is related to the Pacific Decadal Oscillation (PDO). These occur associated with higher frequency of the positive phase of the PDO, resulting in warmer air temperatures, while a negative trend in precipitation occurs during El Nino years, resulting in a decline in open water in the Columbia valley wetlands (as discussed in Lines 478 to 492).

Same point at lines 408 – 409.

Authors: These points reinforce how the peak discharge shifted relative to pre-1928 (eleven days earlier and that the peak flow duration decreased by eleven days), and how the increase of air temperature has impacted snowmelt, therefore, raising the peak discharge and reducing peak season duration (Lines 432 to 435). In addition, the effect of the warmer years with El Niño and a consequence of the reduction of precipitation (Lines 478 to 492).

**Section 3.3**

(line 345) What I see in Figure 10 is not that 'higher peaks appeared more frequently'. Indeed, as you said at lines 407-408, the highest peaks were observed during 1903-1978. I would rephrase sentence at line 345.

Authors: We thank the reviewer for this useful comment. As recommended, we rephrased line 345 (now line 346).

> *Lines 346 – 347: "The frequency analysis revealed that higher peaks appeared more frequently during 1903 – 1978, since the highest peak was 770 $m^3.s^{-1}$, while the lowest peak after 2003 was 305 $m^3.s^{-1}$, which was 233% lower (Figure 10a)."*

**Section 4.3**

I appreciate Figure 12, which provides clearly the seasonal variability of the main hydrological processes. I would include in that figure the percentages of increasing/decreasing of the mentioned variables.

Authors: We thank the reviewer for his/her compliment in Figure 12. As suggested, we included the percentage of increase/decrease in the mentioned landcover.

[Figure]

**Figure 12:** Updated conceptual understanding of the hydrological processes conditions for the UCRW during Spring (a), Summer (b), and late Summer (c).

Overall, I suggest a minor revision of this work. I hope authors will find these comments useful for their research.

Authors: We would like to thank Reviewer #1 for the constructive comments and suggestions on our manuscript.

**Typos**

(line 17) 'land cover trends' instead of 'landcover trends'

Authors: As recommended, in line 17, we changed from "landcover trends" to "land cover trends".

(Line 236) 'it was used' instead of 'was used' (am I right?)

Authors: We thank the reviewer for this comment. As suggested, we changed from "was used" to "it was used".

**Reviewer #2:** The authors present a detailed analysis of the evolving landscape of the Upper Columbia River Wetlands (UCRW) in British Columbia, Canada. Through the application of a Random Forest algorithm to Landsat image archives from 1984 to 2022, this paper methodically examines the impact of changing hydro-climatological variables, including air temperature, river discharge, and water levels, on the wetland's landcover. The study's findings are significant, revealing pronounced changes in regional hydrology and vegetation communities, primarily attributed to climate change. This research provides critical insights into the consequences of climate change on wetland ecosystems and underscores the importance of implementing effective management strategies for their preservation. The study concludes by highlighting the increased vulnerability of these wetland habitats to future climatic and hydrological shifts, which may further diminish their extent.

Authors: Dear Reviewer #02, we would like to thank you very much for the positive feedback and recommendations, as well as all the time spent on our manuscript. Please find a point-by-point answer.

The authors have applied appropriate methods and their approach is grounded in a comprehensive and well-documented review of the relevant literature. The explanation of the methods is detailed and clear. The figures are clear and well-produced. The manuscript is well-written with very minor typos that I was able to catch e.g., whether in line 431.

Authors: Thank you very much for the positive comment. As suggested, we corrected the minor errors over the manuscript, as well as the one you pointed in Line 431 (now in Line 436).

A lot of important information is included in the supplemental materials regarding the RF model evaluation. If possible including more information about the performance of the model and the effect of the different sample sizes on the performance would be great instead of needing to review the supplemental materials.

Authors: We thank the reviewer for this useful comment. Regarding the average performance of the RF model, we showed that in Lines 252 to 254, then, in Line 255 we presented the average performance for the whole year.

Regarding the sample sizes, the years with more high-resolution image availability for training samples (Aerial photographs – 1984 to 1991, 2005, 2007, and 2009; Sentinel 2 – 2016 to 2022; Airborne LiDAR – 2018; UAV LiDAR and geotagged photos – 2022) were the ones where RF performed better (Kappa ≥ 0.8). This illustrates the importance of the sample size in the classification process. We have added a sentence to the Discussion, section 4.1, highlighting this aspect of the classification process.

> *Lines 372 – 374: "Furthermore, the years with supplementary available imagery (1984 to 1991, 2005, 2007, 2009, 2016, 2018, 2022) for additional training and validation were those where RF performed better (average Kappa of 0.84). This illustrates how the increase in high quality training/validation sites improves the land cover classification."*

If the authors can document and release their GEE notebook and data that would be greatly appreciated.

Authors: The GEE notebook and the land cover data that support the findings of this study are actively being used and refined for other related studies. These can be made available from the corresponding author upon request.

In Lines 547 to 549, we inserted the code availability link for Landsat 5 and 8, as well as a statement that the data availability would be provided upon request.

> *Lines 547 – 549: Code availability. Code available in the link:*
> *https://github.com/italosrodrigues/GEE-RF-LC-code*
>
> *Data availability. The land cover data that support the findings of this study are actively being used for other related studies. The data that support the findings of this study are available from the corresponding author upon reasonable request.*

**Editor:** I agree with the reviewers that the manuscript presents a valuable study on trends and changes in landcover and hydro-climatological variables in the Upper Columbia River Wetlands (UCRW). However there are still some aspects that the referees have identified, through a series of comments, that will help improve the manuscript before considering publication. Some of the key comments include:

Authors: Dear Editor, we would like to thank you for the positive feedback and suggestions, as well as the time spent to help us improve our research. Please find point-by-point answers below.

1) The need to clarify some methodological aspects (i.e. on the analysis of peak flow and base flow) as suggested by reviewer #1

Authors: As recommended, we clarified the analysis of peak flow and base flow in section 2.7.

The aim was to demonstrate the temporal variation of the peak flow, which has tended towards earlier occurrence and shorter duration, thus impacting the open water area, and inhibiting the growth of some species in some locations or initiating succession processes in others.

However, we have updated Figure 11 which now shows that the twenty-five-year interval average baseflow discharge hydrograph of the Upper Columbia River has not changed considerably since 1903. In addition, we rephrased a sentence in the Results (section 3.3, Lines 353 to 355) and Discussion (section 4.2, Line 415), adding clear and quantifiable results about the baseflow. We believe that with this addition, it should be clear that the changes in peak flow conditions are of primary interest here.

2) Clarifying information on model evaluation and performance.

Authors: As suggested, we included more information about the model evaluation and performance in the manuscript. Regarding the average performance of the RF model, we showed that in Lines 252 to 254, then, in Line 255 we presented the average performance for the whole year.

We have added a sentence (in Lines 372 – 374) to the Discussion, section 4.1, highlighting the sample sizes, and how the years with more high-resolution image availability for training samples were the ones where RF performed better (Kappa ≥ 0.8). This illustrates the importance of the sample size in the classification process.

3) The inclusion of the GEE notebook (suggested by reviewer#2) would be very important for reproducing the results. Though the authors mention that the GEE notebook is constantly updated, the version used for the paper could be made available as it is useful for reproducing the results of the current paper. Including the GEE and, particularly, the data used will be very important. Please look at the data policy guidelines.

Authors: The GEE notebook code availability link for Landsat 5 and 8 that performed the land cover classification in the UCRW was shared in Lines 547 to 549. The historical land cover data of the UCRW is actively being used for other related studies, however, it can be made available from the corresponding author upon request.

In addition to these main points, both reviews include very useful feedback and several other comments that are extremely valuable (including the suggestion to improve figure 12). I appreciate the authors have posted detailed and positive responses to the comments, which is very important as after resubmission the paper might be returned to the reviewers for further assessment.

Authors: I would like to thank the Editor and the Reviewers for the dedication and work you have dispensed in our manuscript. All recommendations and questions were accepted and addressed, respectively, including the suggestion to improve Figure 12.

Following up on point 3 in the list above, please note that you need to include a statement on code and data availability on the manuscript (https://www.hydrology-and-earth-system-sciences.net/submission.html)

Authors: The final manuscript includes a code statement (Lines 547 and 548) as well as information about data availability (Line 549).

Yours sincerely,

Italo Rodrigues, Christopher Hopkinson, Laura Chasmer, Ryan J. MacDonald, Suzanne E. Bayley, Brian Brisco